# QuantSpec: Self-Speculative Decoding with Hierarchical Quantized KV Cache

**Rishabh Tiwari** [1] [*]  **Haocheng Xi** [1] [*]  **Aditya Tomar** [1] [*]  **Coleman Hooper** [1]  **Sehoon Kim** [1]  **Maxwell Horton** [2]
**Mahyar Najibi** [2]  **Michael W. Mahoney** [1] [3] [4]  **Kurt Keutzer** [1]  **Amir Gholami** [1] [3]

## Abstract

Large Language Models (LLMs) are increasingly being deployed on edge devices for long-context settings, creating a growing need for fast and efficient long-context inference. In these scenarios, the Key-Value (KV) cache is the primary bottleneck in terms of both GPU memory and latency, as the full KV cache must be loaded for each decoding step. While speculative decoding is a widely accepted technique to accelerate autoregressive decoding, existing methods often struggle to achieve significant speedups due to inefficient KV cache optimization strategies and result in low acceptance rates. To address these challenges, we propose a novel self-speculative decoding framework, QuantSpec, where the draft model shares the architecture of the target model but employs a hierarchical 4-bit quantized KV cache and 4-bit quantized weights for acceleration. QuantSpec maintains high acceptance rates (>90%) and reliably provides consistent end-to-end speedups upto $\sim 2.5\times$, outperforming other self-speculative decoding methods that use sparse KV cache for long-context LLM inference. QuantSpec also reduces the memory requirements by $\sim 1.3\times$ compared to these alternatives.

## 1. Introduction

Large Language Models (LLMs) have been widely used in recent years, revolutionizing natural language processing (NLP) and artificial intelligence (AI) applications. As their applications expand, there is a growing demand to deploy LLMs in long-context settings – handling extended text inputs such as document summarization, lengthy conversations, or comprehensive instructions. The model must maintain coherence in such contexts and track intricate de-

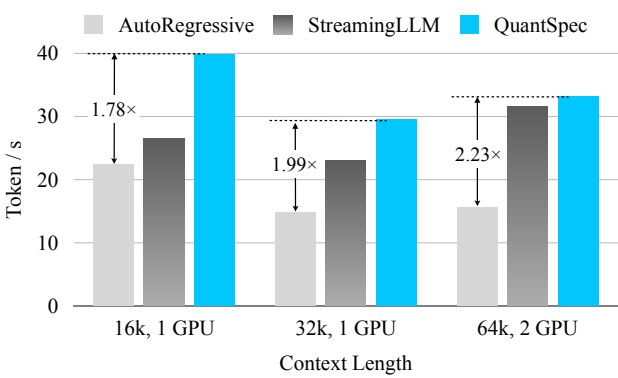

*Figure 1.* Throughput in tokens/sec of various decoding methods. QuantSpec achieves $> 1.78\times$ speedup over the autoregressive baseline across several context lengths. Benchmarked on LWM-Text-Chat-128k.

tails across extended sequences. However, long-context inference presents significant challenges in terms of efficiency and scalability. For example, token eviction (Zhang et al., 2024d; Ge et al., 2023; Liu et al., 2024b) and KV cache quantization (Liu et al., 2024c; Kang et al., 2024; Hooper et al., 2024) have been proposed to improve the efficiency for long-context inference. However, they often entail noticeable degradation in generation quality.

One promising alternative to enhance the efficiency of LLMs while preserving generation quality is speculative decoding (Leviathan et al., 2023; Chen et al., 2023; Kim et al., 2024). This method accelerates inference by using a smaller (draft) model to rapidly generate candidate tokens, and uses the original (target) model to verify these tokens to ensure generation quality. However, the efficient application of speculative decoding in long-context settings has not been thoroughly explored.

Traditional speculative decoding approaches often rely on using smaller models as the draft model in order to minimize the memory-bandwidth overhead of loading the *model weights* of the larger target model. In long-context scenarios, however, the primary bottleneck shifts from model weights to the *KV cache*, which grows linearly with the context length. Additionally, since small models do not usually possess good long-context understanding ability, the acceptance rates of the candidate tokens by the target model drop signifi-

[*]Equal contribution [1]UC Berkeley [2]Apple [3]ICSI [4]LBNL. Correspondence to: Amir Gholami <amirgh@berkeley.edu>.

*Proceedings of the 42$^{nd}$ International Conference on Machine Learning*, Vancouver, Canada. PMLR 267, 2025. Copyright 2025 by the author(s).

cantly, leading to suboptimal speedup. Moreover, traditional speculative decoding methods maintain the KV cache for both the target model and the draft model, causing a large memory footprint. Therefore, finding a solution that both optimizes the KV cache's memory efficiency and improves the acceptance rate within speculative decoding is essential for performant LLMs in long-context applications.

To mitigate these issues and to enable efficient and accurate long-context inference, we propose QuantSpec, a self-speculative decoding method that utilizes 4-bit weights and a 4-bit hierarchical KV cache to speedup long-context inference. In particular we make the following contributions:

- We perform a comprehensive analysis of LLM inference to identify bottlenecks across various context lengths, demonstrating that quantizing the KV cache improves efficiency for long contexts, while quantizing model weights is more beneficial for short contexts (see Section 3.1).

- We introduce a novel hierarchical quantization technique that enables bit-sharing between the target and draft models' KV caches, eliminating the need for additional memory for the draft model (see Section 4.2).

- We propose a double full-precision cache buffer used for storing the most recent KV cache in full precision to improve acceptance rates and also eliminate wasteful quantization and dequantization operations (see Section 4.3).

- We show that using a quantized KV cache leads to better acceptance rates between the target and the draft model, and thus leads to better overall speedups (see Section 5.2).

- We implement custom CUDA kernels for attention with our hierarchical quantized KV cache achieving up to $\sim 2.88\times$ speedups at 4-bit precision relative to FP16 FlashAttention kernels. (see Section 5.2.1)

## 2. Related Work

### 2.1. Efficient Long-Context Inference

An important challenge in optimizing long-context inference lies in reducing memory and computation requirements while retaining high performance on tasks that involve long sequences. Sparse attention mechanisms (Liu et al., 2021; Xiao et al., 2023b; Yao et al., 2024; Tang et al., 2024; Yang et al., 2024; Liu et al., 2024b; Ge et al., 2023; Jiang et al., 2024) have been widely adopted to manage the quadratic complexity of traditional full attention in long contexts. These techniques typically maintain efficiency by dropping non-essential Key-Value (KV) pairs from the cache. Token pruning (Fu et al., 2024) selectively computes the KV

for tokens relevant for next token prediction. KV Prediction (Horton et al., 2024) improves prompt processing time by predicting the KV cache needed for autoregressive generation. Retrieval-augmented generation (Tan et al., 2024; Liu et al., 2024a) enhances the accuracy of language model outputs by combining generative models with external retrieval mechanisms whose context length is very long.

### 2.2. Quantization

Quantization has emerged as a powerful technique to reduce the memory footprint and computational complexity in large-scale neural networks. Weight-only quantization (Lin et al., 2024; Kim et al., 2023a; Shao et al., 2023; Chee et al., 2024) focuses on reducing the precision of model weights to reduce the memory requirements of the model. As models grow larger, the memory footprint of KV caches can become substantial, especially for long input sequences. KV cache quantization (Liu et al., 2024c; Hooper et al., 2024; Kang et al., 2024) addresses this issue by quantizing the key and value caches to enable longer sequence inference.

### 2.3. Speculative Decoding

Speculative decoding has become an important technique for improving the inference efficiency of LLMs (Leviathan et al., 2023; Chen et al., 2023; Kim et al., 2024). It uses a smaller draft model to rapidly generate candidate tokens, which are then verified by a larger target model to ensure correctness. Parallelization in speculative decoding has also been studied to enhance the efficiency by predicting multiple tokens at one time (Cai et al., 2024; Bhendawade et al., 2024; Li et al., 2024b; Chen et al., 2024b). We include additional related works in Appendix B.

**Self-speculative decoding** is the class of speculative decoding methods in which the draft model shares the same architecture as that of target model for better alignment. Recent works like Magicdec (Sadhukhan et al., 2024) and TriForce (Sun et al., 2024) have shown that self-speculation with sparse KV can effectively speedup the draft model in long-context settings, where KV is the main bottleneck. While this design avoids loading the entire KV cache throughout the autoregressive generation process, KV cache sparsification can lead to noticeable performance degradation as evidenced in previous works (Zhang et al., 2024d; Liu et al., 2024b; Ge et al., 2023; Zhou et al., 2024). This can potentially yield a mismatch between the draft and target model's predictions (i.e., lower acceptance rate), which is a critical factor in overall speedup. QuantSpec addresses this limitation by proposing a draft model with a novel hierarchical quantized KV cache, which maintains a higher acceptance rate between the draft and target models, therefore leading to better speedup. Note that our method can be combined with sparse KV methods (Sun et al., 2024;

Sadhukhan et al., 2024) for additional speedup, which we leave for future work.

# 3. LLM Inference Bottlenecks

## 3.1. Arithmetic Intensity

To understand the primary bottlenecks in LLM inference and to motivate our method, we perform a thorough analysis of inference under several different regimes. These regimes include a combination of small versus large batch sizes and short versus long context lengths during both the prefill and decoding stages. We use arithmetic intensity as the central metric in our analysis, where arithmetic intensity is defined as the number of floating point operations (FLOPs) that can be performed per byte loaded from memory, or memory operations (MOPs) (Williams et al., 2009):

$$\text{Arithmetic Intensity} = \frac{\#\,\text{FLOPs}}{\#\,\text{MOPs}}.$$

Arithmetic intensity allows us to classify which regimes of LLM inference are compute-bound or memory-bound and determine appropriate optimizations to improve latency. Compute-bound operations are limited by the hardware's peak FLOP/s (FLOPs per second) performance and benefit from algorithmic improvements that reduce computational complexity (e.g., subquadratic attention). On the other hand, memory-bound operations are limited by the hardware's memory bandwidth (GB/s) and benefit from techniques that optimize memory load-store operations, such as quantizing the weights of a model even if they are later scaled up to a higher precision during computation to preserve accuracy.

For a finer-grained analysis, we break down the major operations in the Transformer into two categories: **linear**, which consists of the weight-to-activation matrix multiplications (i.e., $W_Q, W_K, W_V, W_{out}$, `mlp_up_proj`, `mlp_down_proj`, and the linear classification layer), and **attention**, which consists of the activation-to-activation matrix multiplications (i.e., query $\times$ key and attention weights $\times$ values). Note that the **aggregate** of all Transformer operations includes the above operations as well as non-linear operations like activation functions in the feed-forward network, softmax in the attention mechanism, and layer normalization. Because we are interested in studying the linear and attention operations, we do not explicitly focus on the non-linear operations and classification layer in our asymptotic analysis, although we include them in our final results.

### 3.1.1. ASYMPTOTIC ANALYSIS OF ARITHMETIC INTENSITY FOR PREFILL AND DECODING

During **prefill**, the model weights are only loaded once to process all tokens in the input and generate the first token. Because the context length can range from a couple thousand to hundreds of thousands of tokens, this phase consists of large matrix-matrix multiplications (matmuls) with high arithmetic intensities. Table 1 shows asymptotic analysis of arithmetic intensity for prefill and decoding broken up into linear, attention, and aggregate operations for batch size $B$, sequence length $S_L$, hidden dimension $d$, and a generation length of $k$ tokens. During prefill, the aggregate arithmetic intensity is similar to the arithmetic intensity of the linear projections when $S_L \ll d$ because self-attention is relatively inexpensive for short contexts. Thus the linear projections dominate latency in this regime. However, as the context length increases and $S_L \gg d$, the aggregate arithmetic intensity reflects the arithmetic intensity of attention, which begins to dominate latency since self-attention incurs additional cost with longer context lengths. Note that our analysis assumes the use of FlashAttention (Dao et al., 2022), such that the attention scores matrix which grows on the order of $\mathcal{O}(B \cdot S_L{}^2)$ is never fully materialized, and thus the memory operations for this matrix are limited to $\mathcal{O}(B \cdot S_L)$.

On the other hand, in the **decoding** stage, generating $k$ tokens requires loading and storing the weights and KV cache $k$ times. Since the input at each iteration is a single token per sequence in the batch ($x \in \mathbb{R}^{B \times 1 \times d}$), these operations mainly consist of small matmuls with low arithmetic intensity. For short context lengths where $S_L \ll d$, the aggregate arithmetic intensity for decoding again reflects the arithmetic intensity of the linear projections as loading and storing a small KV cache is relatively inexpensive compared to loading and storing the model weights. However, as the context length grows ($S_L \gg d$), the load-store operations for the large KV cache exacerbate and dominate latency, and the aggregate arithmetic intensity reflects the arithmetic intensity of attention. Ultimately, the aggregate arithmetic intensity for decoding is much lower than that of prefill:

$$\underbrace{\begin{cases} \mathcal{O}(B \cdot S_L), & S_L \ll d \\ \mathcal{O}(S_L), & S_L \gg d \end{cases}}_{\text{prefill}} \gg \underbrace{\begin{cases} \mathcal{O}(B), & S_L \ll d \\ \mathcal{O}(1), & S_L \gg d \end{cases}}_{\text{decode}}.$$

While the aggregate arithmetic intensity for prefill scales proportionally to the context length which can be in the hundreds of thousands, the aggregate arithmetic intensity for decoding does not scale with the context length at all. Moreover, using larger batch sizes only seems to increase the arithmetic intensity for decoding in the short-context setting. For long contexts, decoding has an extremely low arithmetic intensity irrespective of the batch size since every sequence in the batch undergoes self-attention separately and therefore cannot benefit from batching in the same way linear layers do.

*Table 1.* Asymptotic analysis of arithmetic intensity for linear, attention, and aggregate operations under prefill and decoding for batch size $B$, sequence length $S_L$, hidden dimension $d$, and generation length of $k$ tokens.

| **Prefill** | | | |
|---|---|---|---|
| | Linear | Attention | Aggregate |
| FLOPs | $\mathcal{O}(B \cdot S_L \cdot d^2)$ | $\mathcal{O}(B \cdot S_L{}^2 \cdot d)$ | $\mathcal{O}(B \cdot S_L \cdot d^2) + \mathcal{O}(B \cdot S_L{}^2 \cdot d)$ |
| MOPs | $\underbrace{\mathcal{O}(B \cdot S_L \cdot d)}_{\text{activations}} + \underbrace{\mathcal{O}(d^2)}_{\text{weights}}$ | $\underbrace{\mathcal{O}(B \cdot S_L)}_{\text{flash-attn scores}} + \underbrace{\mathcal{O}(B \cdot S_L \cdot d)}_{\text{activations } \{Q, C_K, C_V\}}$ | $\mathcal{O}(B \cdot S_L \cdot d) + \mathcal{O}(d^2)$ |
| Arithmetic Intensity | $\approx \begin{cases} \mathcal{O}(B \cdot S_L), & S_L \ll d \\ \mathcal{O}(d), & S_L \gg d \end{cases}$ | $\approx \begin{cases} \mathcal{O}(S_L), & S_L \ll d \\ \mathcal{O}(S_L), & S_L \gg d \end{cases}$ | $\approx \begin{cases} \mathcal{O}(B \cdot S_L), & S_L \ll d \\ \mathcal{O}(S_L), & S_L \gg d \end{cases}$ |
| **Decode** | | | |
| | Linear | Attention | Aggregate |
| FLOPs | $\mathcal{O}(k \cdot B \cdot d^2)$ | $\mathcal{O}(k \cdot B \cdot S_L \cdot d)$ | $\mathcal{O}(k \cdot B \cdot d^2) + \mathcal{O}(k \cdot B \cdot S_L \cdot d)$ |
| MOPs | $\underbrace{\mathcal{O}(k \cdot B \cdot d)}_{\text{activations}} + \underbrace{\mathcal{O}(k \cdot d^2)}_{\text{weights}}$ | $\underbrace{\mathcal{O}(k \cdot B \cdot S_L)}_{\text{attention scores}} + \underbrace{\mathcal{O}(k \cdot B \cdot S_L \cdot d)}_{\text{activations } \{C_K, C_V\}}$ | $\mathcal{O}(k \cdot d^2) + \mathcal{O}(k \cdot B \cdot S_L \cdot d)$ |
| Arithmetic Intensity | $\approx \begin{cases} \mathcal{O}(B), & S_L \ll d \\ \mathcal{O}(B), & S_L \gg d \end{cases}$ | $\approx \begin{cases} \mathcal{O}(1), & S_L \ll d \\ \mathcal{O}(1), & S_L \gg d \end{cases}$ | $\approx \begin{cases} \mathcal{O}(B), & S_L \ll d \\ \mathcal{O}(1), & S_L \gg d \end{cases}$ |

### 3.1.2. COMPUTE VERSUS MEMORY-BOUND REGIMES

The asymptotic analysis suggests that in general, decoding suffers from low arithmetic intensities compared to prefill in all regimes. However, to decide which optimizations will most effectively improve latency, all regimes must be classified as either compute-bound or memory-bound. Whether an operation is compute or memory-bound depends on the hardware it is being run on as well as the magnitude of the arithmetic intensity achieved by the operation.

We utilize an analytical roofline model (Williams et al., 2009; Kim et al., 2023a;b) to help determine which regimes are compute or memory-bound in a practical inference setting. The roofline model defines a *ridge point* which is calculated as

$$\frac{\text{peak compute performance (FLOP/s)}}{\text{peak memory-BW (GB/s)}}.$$

Note that the ridge point has the same units as arithmetic intensity (FLOPs/byte). In the roofline model, any operation with an arithmetic intensity smaller than the ridge point is memory-bound, and any operation with an arithmetic intensity greater than the ridge point is compute-bound. For our analysis, we extrapolate this to a *ridge plane* and use hardware specifications for an NVIDIA A6000 GPU to study inference for the Llama-2-7B model in 16 bit precision.

For optimizing speculative decoding, we specifically focus on the decoding phase, although we include results for prefill in Appendix C.1. Figure 2 shows the arithmetic intensity for generating 1k tokens at different context lengths and batch sizes for the Linear/Attention components as well as the aggregate arithmetic intensity. To decide the ideal quantization strategy for different regimes, we consider the aggregate arithmetic intensity, which is colored by the percentage of the total latency taken up by attention and provides a complete view of decoding in all regimes. Based on these results, we can clearly see that in the small batch + short context regime, the memory operations for the linear projections dominate latency, so **weight quantization** could provide considerable speedup in this regime. In the small batch + long context, large batch + short context, and large batch + long context regimes, attention dominates latency due to the expensive load-store operations for the large KV cache. **KV cache quantization** could help provide performance improvements in these regimes. In the small batch + medium context and short context + medium batch regimes, the linear and attention operations are approximately equivalent in their contributions to total latency. Thus, both **weight and KV cache quantization** are ideal here.

## 4. QuantSpec

### 4.1. Overview of QuantSpec

In this section, we introduce QuantSpec, a self-speculative decoding framework designed to accelerate both short- and long-context generation by quantizing the model weights and KV cache into INT4 precision. We begin by noting that self-speculative decoding is particularly well-suited

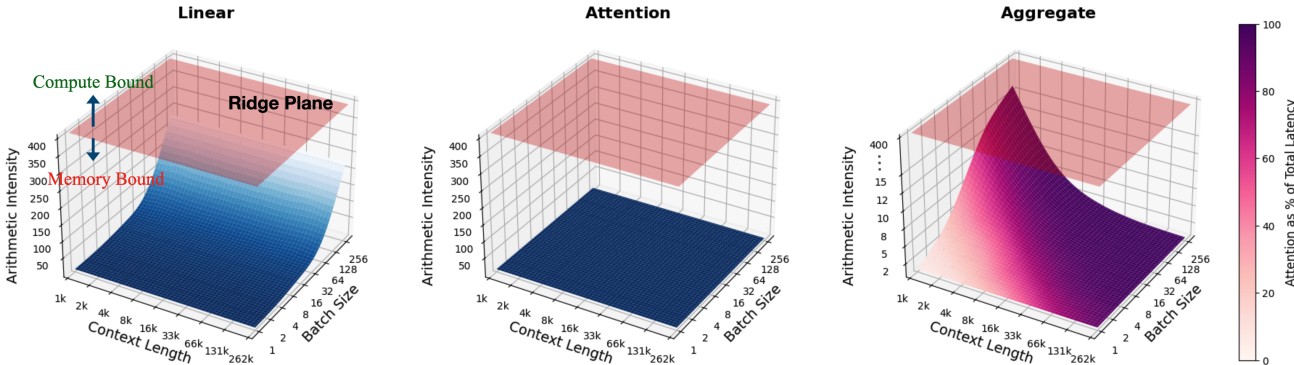

*Figure 2.* Breakdown of how arithmetic intensity changes during decoding as the context length and batch size are scaled logarithmically for linear, attention, and aggregate operations. All regimes lie below the ridge plane and thus are memory-bound. The ridge plane is calculated for an NVIDIA A6000 GPU. The colors for the linear and attention surface plots simply represent the magnitude of the arithmetic intensity. The aggregate plot is colored by attention's runtime as a percentage of the total latency. Prefill results in Appendix C.1.

for long-context generation, as the draft model shares the same architecture as the target model. This architectural alignment improves both the acceptance rate and the model's ability to handle long contexts effectively. However, a naive implementation of self-speculative decoding (e.g. based on sparse KV) would require maintaining a separate, fully quantized copy of the KV cache, leading to inefficiencies in memory usage and computational overhead.

To address this limitation, QuantSpec introduces a novel hierarchical KV cache design, which we discuss in detail in Section 4.2. This design enables dynamic switching between INT4 and INT8 representations of the KV cache without the overhead of on-the-fly quantization. By eliminating redundancy between the draft and target models' KV caches, our method significantly reduces the total memory footprint while preserving efficiency. We also address the inefficient combination of conventional quantization strategies with the reject-and-revert-back mechanism specific to speculative decoding methods by proposing a full-precision KV cache buffer in QuantSpec. As we further explain in Section 4.3, this helps achieve high acceptance rates for the draft model and thus results in greater end-to-end speedup.

### 4.2. Hierarchical KV Cache

We propose a 4-bit hierarchical KV cache wherein we strategically structure each tensor's representation such that the draft and target models are able to dynamically reconstruct their KV cache without any on-the-fly quantization overhead. Firstly, we observe that using an INT8 KV cache for the target model is comparable in terms of accuracy and performance with the same target model using an FP16 KV cache. To demonstrate this, we conduct a perplexity analysis for Llama-2-7B on the WikiText-2 (Merity et al., 2016) and C4 (Raffel et al., 2020) datasets in Table 2, which shows that the target model with an INT8 KV cache maintains com-

petitve generation quality with respect to the FP16 baseline while using half the KV cache's memory.

| | Datasets | |
|---|---|---|
| KV Cache | WikiText2 | C4 |
| FP16 (Baseline) | 6.4595 | 7.2617 |
| INT8 (QuantSpec target) | 6.4696 | 7.2620 |

*Table 2.* Perplexity evaluations of Llama-2-7B with FP16, INT8 with group size = 128, residual length = 256 on different datasets.

Having observed this, we further note that an INT8 KV cache can be represented as an INT4 KV cache plus its INT4 residual. This works by decomposing an INT8 value into two INT4 components corresponding to its first and second 4-bit segments, which we call the upper and lower 4-bits. This effectively allows us to use a hierarchical design to represent the KV cache of the draft model in INT4 and the target model in INT8 at the same time, removing the need to store a separate INT4 copy.

Our method is visualized in Figure 3. During prefill, QuantSpec quantizes the FP16 KV cache to form the upper and lower INT4 representations. To obtain the upper 4-bits $C_U^{\text{INT4}}$ and lower 4-bits $C_L^{\text{INT4}}$ values, we first calculate $C_U^{\text{INT4}}$, then quantize the quantization error $E_U^{\text{INT4}}$ to get $C_U^{\text{INT4}}$. $C_L^{\text{INT4}} \in [0, 15]$ uses asymmetric and round-to-nearest quantization. Since the distribution of $E_U^{\text{INT4}}$ is symmetric and has an expectation close to zero, for $C_L^{\text{INT4}} \in [-8, 7]$ we use symmetric and round-to-nearest quantization to better match the distribution of errors.

Then during decoding, when using the draft model to generate candidate tokens, we only load the upper 4-bit representation in our kernel and dequantize it for inference. When verifying the drafted tokens using the target model, we utilize both the upper and lower 4-bit representations to reconstruct

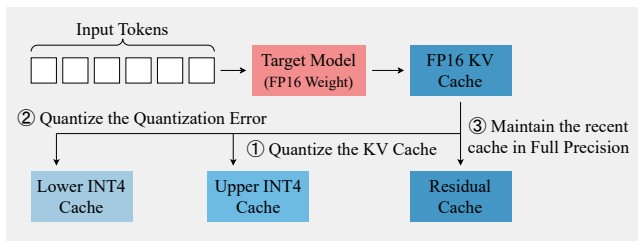

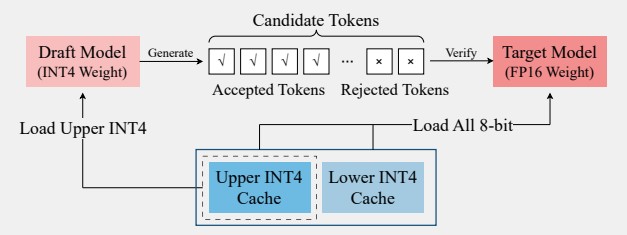

(a) During prefill, our hierarchical KV cache quantizes the FP16 KV cache to get the upper and lower INT4 representations.

(b) During decoding, our hierarchical KV Cache loads the Upper INT4 KV Cache for the draft model and loads all 8-bits for the target model.

*Figure 3.* How our Hierarchical KV Cache works in the speculative decoding setting.

the KV cache in the higher INT8 precision. To represent the INT8 KV cache $C^{\text{INT8}}$ as the upper INT4 KV cache $C_U^{\text{INT4}}$ and the lower INT4 cache $C_L^{\text{INT4}}$, the INT8 KV cache can be expressed as $C^{\text{INT8}} = 2^4 C_U^{\text{INT4}} + C_L^{\text{INT4}}$, where we multiply by $2^4$ to align their represented values. The asymmetric quantization for the KV cache can be represented as $C^{\text{FP32}} = C^{\text{INT8}} S^{\text{INT8}} + Z^{\text{INT8}}$, where $S^{\text{INT8}}$ is the scaling factor, $Z^{\text{INT8}}$ is the zero point, and $C^{\text{INT8}} \in [0, 2^8 - 1]$. In this scenario, its 4-bit representation can be viewed as

$$C^{\text{FP32}} = (2^4 C_U^{\text{INT4}} + C_L^{\text{INT4}}) S^{\text{INT8}} + Z^{\text{INT8}}$$
$$= C_U^{\text{INT4}} S^{\text{INT4}} + C_L^{\text{INT4}} \frac{S^{\text{INT4}}}{2^4} + Z^{\text{INT4}},$$

where $Z^{\text{INT4}} = Z^{\text{INT8}}, \ S^{\text{INT4}} = 2^4 S^{\text{INT8}}$.

### 4.3. KV Cache with Double Full Precision Buffer

#### 4.3.1. CHALLENGES WITH KV CACHE QUANTIZATION AND SPECULATIVE DECODING

The key and value caches have each been found to exhibit unique characteristics indicating that they should be quantized with different strategies (Liu et al., 2024c). Specifically, quantizing the key cache along the channel axis and quantizing the value cache along the token axis minimizes quantization error (as shown in Table 3), and therefore leads to a higher acceptance rate in speculative decoding. We apply *asymmetric quantization* and *per-group quantization* to both the key and value caches in INT4 precision, and we set the group size $G$ to be equal to the head dimension to reduce overhead. These quantization techniques are illustrated in Appendix D Figure 8.

However, these quantization strategies for the key and value caches pose efficiency challenges when combined with speculative decoding. Regarding the value cache wherein the values are quantized along the token axis, the naive strategy of directly quantizing newly generated tokens at each decoding step is expensive, as it introduces high computational overhead that occurs very frequently. Moreoever, regarding the key cache for which we apply quantization along the channel axis, the naive approach is to store multiple tokens

|              |              | Key Cache |              |
|--------------|--------------|-----------|--------------|
| Value Cache  |              | token-wise | channel-wise |
| token-wise   |              | 6.587     | **6.507**    |
| channel-wise |              | 7.041     | 6.911        |

*Table 3.* Perplexity of Llama-2-7B on WikiText-2 dataset with different quantization strategies. Group size $G = 128$. Channel-wise quantization for key cache and token-wise quantization for value cache gives the best performance.

in full precision until they equal the quantization group size, and then quantize them. However, since the KV cache for the most recent tokens is no longer preserved in full precision after quantization, this strategy adversely affects the acceptance rate, thus reducing the effectiveness of speculative decoding. Moreover, since speculative decoding may result in frequent rollbacks due to the target model rejecting the draft tokens, the quantized KV cache for the rejected tokens need to be discarded and replaced with new tokens in the quantization group. This leads to repeated quantization and dequantization, slowing down the decoding process.

#### 4.3.2. ADAPT TO SPECULATIVE DECODING USING FULL PRECISION BUFFER

To enhance efficiency and ensure compatibility with speculative decoding, we propose maintaining a *double full-precision buffer* of size $2G$, where $G$ is the quantization group size. This buffer is divided into two equal parts: $C_{F_1}$ and $C_{F_2}$, each of size $G$. During prefill, we quantize the input tokens in batches of $G$ while ensuring that at least $G$ but no more than $2G$ of the most recent tokens remain in full precision. This ensures that $C_{F_1}$ is always filled. In the decoding stage, newly generated tokens are stored in full precision in the second buffer, $C_{F_2}$. Once the full-precision buffer reaches its maximum capacity of $2G$, we wait for the target model to verify the generated tokens. If any tokens are rejected, we first remove the corresponding full-precision KV cache entries. Then, we quantize $C_{F_1}$ and append it to the quantized KV cache. We then move

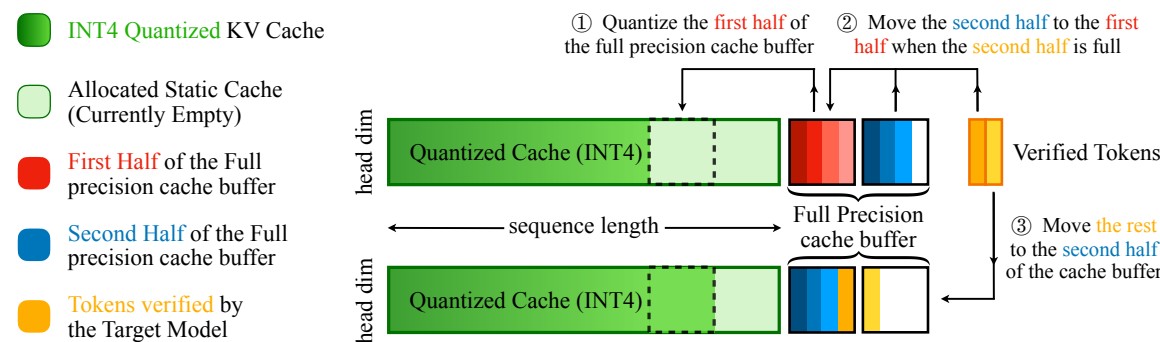

*Figure 4.* Our KV cache with 2 full precision cache buffers for recent KV cache.

$C_{F_2}$ to $C_{F_1}$, which fully occupies $C_{F_1}$ while leaving $C_{F_2}$ empty and ready for tokens generated in future decoding steps. This whole process is visualized in Figure 4.

Using this design, we ensure that (1) at every step $C_{F_1}$ is always filled, so there are at least recent $G$ tokens kept in full precision, which is beneficial for the acceptance rate. (2) Quantization and KV cache movement will only happen every $G$ decoding steps, which significantly reduces the overhead. (3) The design is compatible with speculative decoding since we can discard the KV cache for rejected tokens very flexibly by only operating on the second full-precision buffer $C_{F_2}$ and without needing extra quantize and dequantize operations. We also show that our method is fully compatible with FlashDecoding in Appendix E.

### 4.3.3. SUMMARY

In summary, QuantSpec allows the draft and target models to share the same architecture in a self-speculative decoding manner, ensuring greater consistency between drafting and verification as opposed to traditional big-little speculative decoding methods. Our approach is mainly designed for long-context scenarios, where efficient KV cache management is critical, but it also supports short contexts where using weight quantization becomes more critical. We quantize the KV cache using our hierarchical INT4 design and use a double full-precision cache buffer for higher acceptance rates and flexibility with speculative decoding. Then in the decoding stage, when generating draft tokens, we only load the upper 4-bit of the KV cache and achieve speedup by significantly reducing the memory load/store operations. When verifying these draft tokens, we load both the upper and lower 4-bit KV cache representations and dequantize them into their INT8 representation to achieve performance that is comparable with an FP16 KV cache. If the full-precision buffer is saturated, after verification we quantize and clear one-half of the full-precision buffer to prepare for the next round of generation. The whole algorithm can be visualized in Figure 3 (and Algorithm 1 in Appendix).

## 5. Evaluation

In this section, we evaluate the performance of QuantSpec across multiple datasets and context lengths. Our evaluation focuses on three key dimensions: (1) the acceptance ratio between the draft and target models, (2) GPU memory consumption, and (3) end-to-end serving speedup. We begin by presenting a detailed benchmarking of acceptance rate, memory usage, and end-to-end speedup across different datasets. Then, we highlight the performance gains achieved by our custom kernels for quantized KV cache. Finally, We also present an extensive ablation study focusing on the contribution of weight versus KV cache quantization to the final speed-up.

### 5.1. Setup

All experiments are performed on a node equipped with 8 NVIDIA RTX A6000 GPUs. We evaluate QuantSpec using long-context variants of LLaMA-2 and LWM models as target models. For benchmarking decoding speedup, we use PG-19 (Rae et al., 2019), (an open-vocabulary language modeling benchmark derived from books) and two long context summarization datasets, namely ∞BENCH Sum (Zhang et al., 2024c; Yen et al., 2024) and Multi-LexSum (Shen et al., 2022; Yen et al., 2024). More details about the datasets are provided in Appendix F. Following Sadhukhan et al. (2024), we compare against two recent sparse KV-based self-speculative decoding baselines: StreamingLLM (Sadhukhan et al., 2024; Xiao et al., 2023a) and SnapKV (Sadhukhan et al., 2024; Li et al., 2024a). To ensure a fair comparison, the draft KV budget for the baselines is set to one-fourth of the context length, matching our 4-bit quantized KV cache. We fix the quantization group size at 128, the residual length $R$ for the KV cache at 256, and limit the number of output tokens to 90. The optimal speculation length $\gamma$ for each dataset is determined through a hyperparameter search for each dataset-model pair. Details of the hyperparameter search are provided in Appendix G.

*Table 4.* Efficiency result of QuantSpec on Llama-2-7B-32K-Instruct and LWM-Text-Chat-128k. We benchmark on multiple context length settings, ranging from 4k to 128k for batch size 1, and take their average across 10 different examples. Speedup ratio is compared with autoregressive generation of the target model (AR). We compare with Sparsity-based self-speculative baselines that use StreamLLM and SnapKV to quantize the KV cache. Acceptance rate of each method is shown for the respective optimal $\gamma$. We outperform these baselines and achieve a maximum of 2.49× speedup compared with autoregressive generation.

**Llama-2-7B-32K-Instruct**

| Dataset | Context Length | # GPUs | Method | Acceptance Rate (%) ↑ | Peak GPU Memory (GB) ↓ | Speedup (× AR) ↑ |
|---|---|---|---|---|---|---|
| PG19 | 4k | 1 | StreamingLLM | 88.87 | 15.20 | 1.13 |
| | | | SnapKV | 93.59 | 15.39 | 1.17 |
| | | | QuantSpec | 92.46 | 16.26 | **1.35** |
| | 8k | 1 | StreamingLLM | 90.31 | 17.75 | 1.27 |
| | | | SnapKV | 94.39 | 18.10 | 1.31 |
| | | | QuantSpec | 89.88 | 17.66 | **1.44** |
| Multi-LexSum | 8k | 1 | StreamingLLM | 90.78 | 17.75 | 1.27 |
| | | | SnapKV | 55.55 | 18.10 | 1.01 |
| | | | QuantSpec | 91.23 | 17.66 | **1.61** |
| | 32k | 1 | StreamingLLM | 91.19 | 32.61 | 1.84 |
| | | | SnapKV | 72.54 | 33.96 | 1.63 |
| | | | QuantSpec | 91.16 | 25.84 | **2.08** |

**LWM-Text-Chat-128k**

| Dataset | Context Length | # GPUs | Method | Acceptance Rate (%) ↑ | Peak GPU Memory (GB) ↓ | Speedup (× AR) ↑ |
|---|---|---|---|---|---|---|
| ∞BENCH Sum | 16k | 1 | StreamingLLM | 87.41 | 22.63 | 1.49 |
| | | | SnapKV | 88.61 | 23.33 | 1.50 |
| | | | QuantSpec | 91.64 | 20.36 | **1.78** |
| | 32k | 1 | StreamingLLM | 85.38 | 32.63 | 1.15 |
| | | | SnapKV | 84.55 | 33.98 | 1.78 |
| | | | QuantSpec | 90.99 | 25.86 | **1.99** |
| Multi-LexSum | 64k | 2 | StreamingLLM | 78.95 | 52.96 | 2.16 |
| | | | SnapKV | 82.79 | 55.66 | 2.11 |
| | | | QuantSpec | 92.18 | 38.18 | **2.23** |
| | 128k | 2 | StreamingLLM | - | OOM | - |
| | | | SnapKV | - | OOM | - |
| | | | QuantSpec | 94.31 | 61.22 | **2.49** |

## 5.2. Speedup Evaluation

Table 4 shows the acceptance rate, GPU memory required, and speedup achieved compared to autoregressive decoding. We observe that QuantSpec provides consistently better speedups for all context lengths. For short and medium context lengths (e.g. 8k and 32k prompt length), QuantSpec achieves ∼1.61× to ∼2.08× speedups respectively on the Multi-LexSum dataset. For longer context lengths (e.g. 128k), our speedups are even greater, up to ∼**2.49×**, all while using lower GPU memory than the baselines. We also see that acceptance rates of QuantSpec are considerably higher than the baselines for summarization tasks (refer to Appendix H for detailed comparison); this shows that for such tasks where the whole context is important, sparse KV cache methods are much more lossy, whereas quantization preserves most of the information in the context. Consequently, QuantSpec proves to be a more reliable choice, delivering consistent speedups across varying context lengths and query complexities.

### 5.2.1. KERNEL SPEEDUPS

In Table 5 we show the speedup achieved using our custom attention kernel that makes use of quantized KV cache versus the standard FP16 FlashAttention kernels. For a context length of 128k, our INT4 attention kernel is ∼ 2.88× faster than the standard FlashAttention kernel.

*Table 5.* Latency benchmark of our custom attention kernels for calculating attention with quantized hierarchical KV cache. QuantSpec INT4 refers to only loading the upper-4-bit, QuantSpec INT8 refers to loading both the upper and lower 4-bit. Benchmarked on kernel level.

| | Context Length | |
|---|---|---|
| Kernels | 64k | 256k |
| FlashAttention (FP16) | 3.07 ms | 6.16 ms |
| QuantSpec INT 8 | 1.08 ms (1.44x) | 4.06 ms (1.51x) |
| QuantSpec INT 4 | 0.54 ms (2.88x) | 2.15 ms (2.86x) |

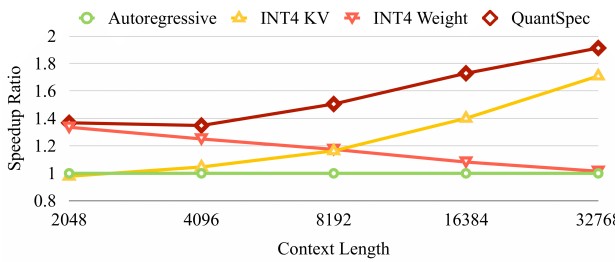

*Figure 5.* Speedup ratio of QuantSpec compared to autoregressive baseline as we scale the context length. We report QuantSpec with KV cache-only quantization, weight-only quantization, and both. Benchmarked on Llama-2-7B-32k-Instruct using PG-19.

### 5.3. Ablation Results

We present an extensive ablation study of QuantSpec focusing on the contribution of weight versus KV cache quantization to the final speed-up.

**Weight versus KV Quantization**: Figure 5 illustrates the speedup ratio of QuantSpec compared to autoregressive baseline as context length increases. The figure benchmarks QuantSpec with KV cache-only quantization, weight-only quantization, and both. The results are aligned with the analysis done in Section 3.1, showing that for short contexts most of the speedup comes from quantizing weights, for medium length prompts both weight and KV cache quantization contribute to the final speedup, and KV cache quantization is most effective for long contexts.

## 6. Conclusions

In this paper, we have introduced a novel approach to enhance the efficiency and scalability of Large Language Models (LLMs) in long-context settings through quantized speculative decoding. Our method addresses the increasing memory and computational demands by optimizing the Key-Value (KV) cache operations, which become a significant bottleneck as the context length grows. We propose a double full-precision cache buffer to resolve conflicts between per-group quantization and speculative decoding. Our comprehensive approach shows that by integrating advanced quantization techniques with speculative decoding, it is possible to significantly improve processing speed without compromising the accuracy and performance of LLMs. This work paves the way for more scalable and effective deployment of LLMs in applications that require extensive contextual understanding, offering a robust solution to the challenges posed by long-context settings.

## Impact Statement

This paper presents work whose goal is to advance the field of Machine Learning. There are many potential societal consequences of our work, none which we feel must be specifically highlighted here.

## Acknowledgments

We acknowledge gracious support from Apple team, as well as Nvidia for providing GPU hardware. We also appreciate the support from Microsoft through their Accelerating Foundation Model Research. Furthermore, we appreciate support from Google Cloud, the Google TRC team, and specifically Jonathan Caton, Divy Thakkar, and Prof. David Patterson. Prof. Keutzer's lab is sponsored by the Intel corporation, Intel One-API, Intel VLAB team, the Intel One-API center of excellence, as well as funding through BDD, BAIR, and Furiosa. We appreciate great feedback and support from Ellick Chan, Saurabh Tangri, Andres Rodriguez, and Kittur Ganesh. Sehoon Kim and Suhong Moon would like to acknowledge the support from the Korea Foundation for Advanced Studies (KFAS). MWM would like to acknowledge DARPA, DOE, NSF, and ONR for providing partial support of this work. Our conclusions do not necessarily reflect the position or the policy of our sponsors, and no official endorsement should be inferred.

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

# Appendix

## A. Attention Module's Inference Workflow

The inference of LLMs can be divided into 2 parts: the prefill stage and the decoding stage. In the **prefill stage**, for the input sequence $X \in \mathbb{R}^{B \times S_L \times d}$, the KV cache update rule can be calculated as

$$Q = XW_Q, \quad C_K = XW_K, \quad C_V = XW_V,$$

where we denote the query, key, and value weight matrices as $W_Q, W_K, W_V \in \mathbb{R}^{d \times d}$ and denote the key and value caches as $C_K$ and $C_V$ respectively. $B$ refers to batch size, $S_L$ refers to sequence length, and $d$ refers to hidden size. We then calculate the multi-head attention (MHA) as:

$$O = \text{MultiHeadAttn}(Q, \ C_K, \ C_V).$$

In the **decode stage**, for input token $x \in \mathbb{R}^{B \times 1 \times d}$, we first calculate the query, key, and value of the current token:

$$q = xW_Q, \quad c_k = xW_K, \quad c_v = xW_V,$$

then concatenate the KV cache with the current token's key and value to update the KV cache:

$$C_K = \text{concat}(C_K, c_k), \quad C_V = \text{concat}(C_V, c_v).$$

Then, the multi-head attention output is calculated:

$$O = \text{MultiHeadAttn}(q, \ C_K, \ C_V).$$

## B. More Related Works

We list some related works that we find interesting, but can not elaborate on in the related works section due to space limitations.

**Efficient Long Context Inference**   Some research maintains the full key-value pairs but dynamically loads them from high-bandwidth memory (Yang et al., 2024; Tang et al., 2024), and usually achieves higher performance at the cost of higher memory consumption. Shared KV cache across tokens (Nawrot et al., 2024) and layers (Brandon et al., 2024) provides a new way to reduce the KV cache budget through sharing.

**Quantization**   Any Precision representation (Park et al., 2024) incorporates multiple precision levels (e.g., INT2, INT4, and INT8) within a single representation, eliminating the need to store separate KV caches for each precision and allowing the framework to dynamically select the optimal precision based on the complexity of the task. Training quantization (Peng et al., 2023; Xi et al., 2024b; Fishman et al., 2024; Xi et al., 2024a) reduces the bit precision of various model parameters, gradients, and activations to accelerate training. Attention quantization (Chen et al., 2024a; Zhang et al., 2024b;a; Weng et al., 2020; Shah et al., 2024) reduces the computational overhead associated with attention computations, which becomes dominant in the prefill stage of the long-context inference setting.

**Speculative Decoding**   Zhao et al., (Zhao et al., 2024) explored complementary quantization schemes in speculative decoding with QSpec, enhancing efficiency without significant performance degradation. Sirius (Zhou et al., 2024) finds that contextual sparsity will lead to poor performance under the speculative decoding setting since the model performance is degraded, and thus it cannot accelerate LLM inference.

## C. Additional LLM Inference Bottlenecks Analysis

### C.1. Prefill Arithmetic Intensity Analysis

Keeping in line with the asymptotic analysis in Table 1, the arithmetic intensity of attention during prefill does not scale with batch size at all, as attention is unable to benefit from batching in the same way that linear operations do. Moreover,

for long contexts, attention entirely dominates the linear operations due to the quadratic nature of self-attention. For short contexts however, this quadratic cost is relatively inexpensive when compared to the linear operations. As shown in Figure 6, the arithmetic intensity for all prefill operations in all regimes is above the ridge plane, which means that prefill is entirely compute-bound.

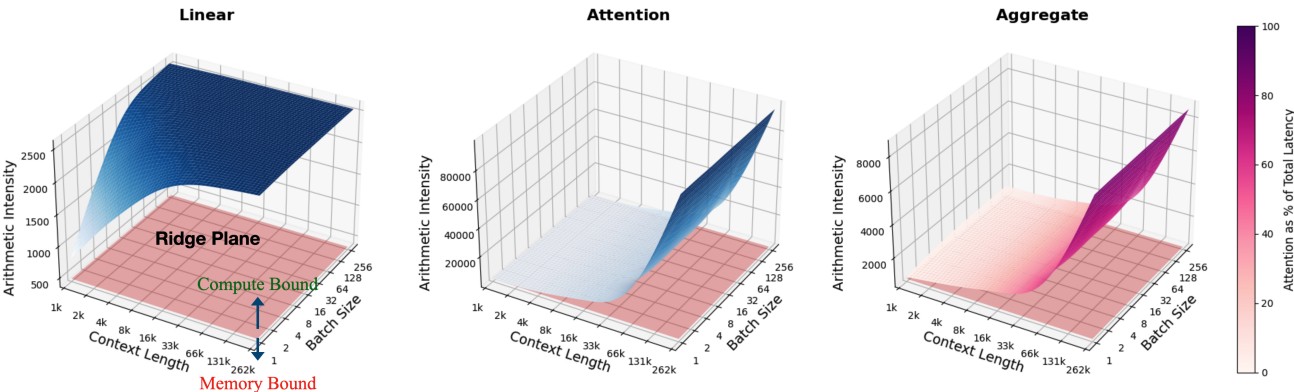

*Figure 6.* During prefill, all regimes lie above the ridge plane and thus are compute-bound.

### C.2. Modern GPU Hardware VRAM Size Constraints

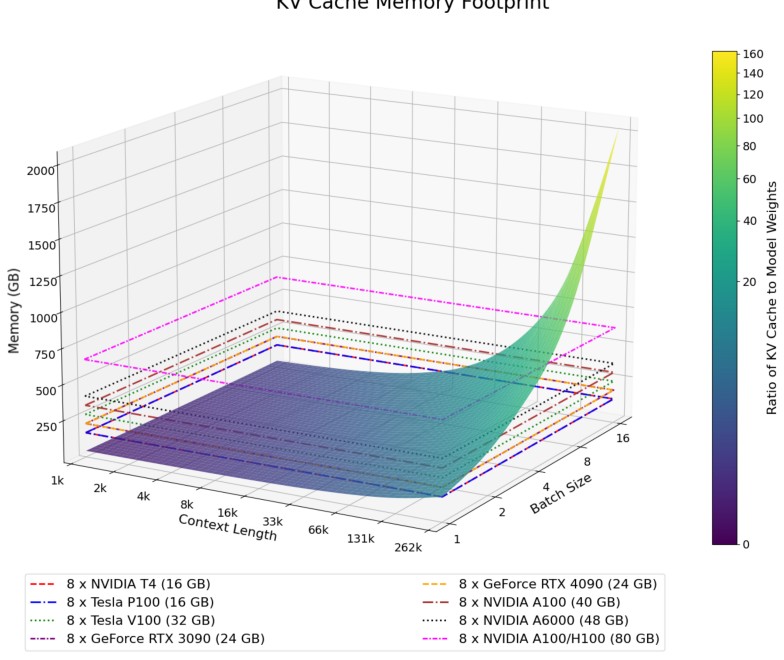

*Figure 7.* KV cache memory usage by Llama-2-7B on a single node (8 GPUs) as context length and batch size are scaled logarithmically. The surface plot's color represents the ratio of KV cache memory to the model weights memory. The dotted-lines represent GPU DRAM capacities for several different GPUs. At ($B = 16, S_L = 262$k), the KV cache takes up 160x more memory than the model weights.

The relatively higher linear arithmetic intensities observed in decoding for batch sizes greater than 8 in Figure 2 are misleading due to the limited VRAM sizes in modern GPUs. As shown in Figure 7, the size of the KV cache for a Llama-2-7B model exceeds the total VRAM capacities of a single node equipped with 8 A100/H100 GPUs with 80 GB of memory each. This means that simply scaling the batch size for decoding will not translate the memory-bound nature of generation to being compute-bound.

# D. Quantization Strategies for KV cache

Here we provide a visualization of our quantization scheme in Figure 8. We apply asymmetric quantization for both the keys and values cache, and apply channel-wise quantization to the key cache and apply token-wise quantization to the value cache.

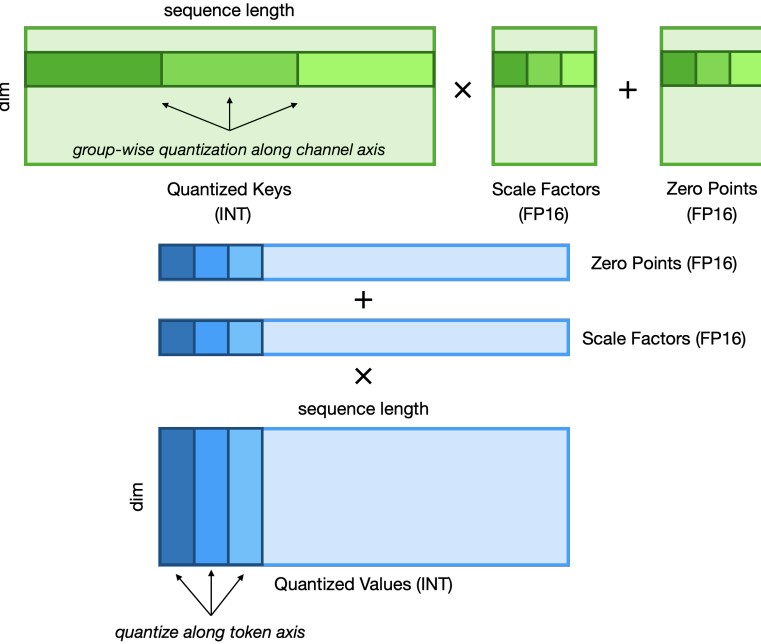

*Figure 8.* We apply asymmetric and per-group quantization for both the key cache and value cache, along the channel axis and token axis, respectively. This figure describes how it works when only the upper-4 bit cache is applied.

# E. Compatibility with Flash Decoding

Our full-precision buffer design, as shown in Figure 4, is fully compatible with Flash Decoding (Dao et al., 2023), a fast attention implementation available for the decoding stage. In this setup, the quantized section divides naturally into several chunks, aligning perfectly with the structure of Flash Decoding. For the full-precision buffer, given its upper bound length of $2G$, it can be processed independently with minimal overhead. This full-precision segment can be treated as an additional chunk, which can then be summed with the quantized segments and seamlessly integrated into the Flash Decoding algorithm.

# F. Details about the Datasets Used

We provide an overview of the datasets used in our experiments, highlighting their key characteristics.

- **WikiText-2** (Merity et al., 2016): WikiText-2 is a widely used dataset for language modeling. It is a subset of the larger WikiText dataset and consists of high-quality, clean, and well-structured English text extracted from Wikipedia articles.

- **C4** (Raffel et al., 2020): C4 is a large scale web-crawled language modelling dataset mostly used for pretraining LLMs.

- **PG-19** (Rae et al., 2019): It is a dataset of books from Project Gutenberg, designed for long-context language modeling.

- **∞BENCH Sum** (Zhang et al., 2024c): InfiniteBench benchmark is tailored for evaluating the capabilities of language models to process, understand, and reason over super long contexts. We used one of its summarization datasets where the task is to summarize a fake book created by core entity substitution. The average length of input prompt is ∼171k.

- **Multi-LexSum** (Shen et al., 2022): Multi-LexSum is a multi-doc summarization dataset for civil rights litigation lawsuits. The average length of prompt in this dataset is ∼90k.

# G. Hyperparameter Search

Here we present details about the hyperparameter search done to select optimal $\gamma$ for each experiment. We search $\gamma$ for each dataset and method pair using a prompt length of 8192 and use the same value for all other context length experiments. Table 6 shows the results of the search. We find that sparse-based methods achieves a maximum performance when $\gamma$ equals to 1, while our quantization-based method usually achieves the best performance with a larger $\gamma$, such as 4 or 6.

*Table 6.* Hyperparameter Search for Llama-2-7B-32K and LWM-Text-Chat-128k models on PG19, Multi-LexSum, and ∞BENCH Sum datasets. Context length is kept as 8k.

(a) Llama-2-7B-32k on PG19

| Method | $\gamma$ | Acceptance Rate ↑ | Speedup ↑ |
|---|---|---|---|
| StreamingLLM | 1 | 90.78 | **39.1** |
| | 2 | 89.42 | 38.5 |
| | 3 | 90.21 | 38.66 |
| SnapKV | 1 | 94.39 | **40.34** |
| | 2 | 91.03 | 39.38 |
| | 3 | 91.84 | 39.28 |
| QuantSpec | 1 | 91.88 | 41.52 |
| | 2 | 89.88 | **44.51** |
| | 4 | 83.17 | 43.84 |
| | 6 | 77.07 | 41.88 |

(b) Llama-2-7B-32k on Multi-LexSum

| Method | $\gamma$ | Acceptance Rate ↑ | Speedup ↑ |
|---|---|---|---|
| StreamingLLM | 1 | 90.78 | **39.17** |
| | 2 | 86.82 | 37.84 |
| | 3 | 83.29 | 36 |
| SnapKV | 1 | 55.55 | **31.05** |
| | 2 | 43.96 | 24.39 |
| | 3 | 36.61 | 19.78 |
| QuantSpec | 1 | 96.58 | 42.83 |
| | 2 | 96.61 | 47.51 |
| | 4 | 95.59 | 49.47 |
| | 6 | 91.23 | **49.62** |

(c) LWM-Text-Chat-128k on ∞BENCH Sum

| Method | $\gamma$ | Acceptance Rate ↑ | Speedup ↑ |
|---|---|---|---|
| StreamingLLM | 1 | 81.79 | **37.17** |
| | 2 | 74.86 | 34.33 |
| | 3 | 64.48 | 30.14 |
| SnapKV | 1 | 85.55 | **38.27** |
| | 2 | 82.92 | 36.97 |
| | 3 | 77.13 | 34.40 |
| QuantSpec | 1 | 93.83 | 42.01 |
| | 2 | 94.38 | 46.74 |
| | 4 | 90.33 | **47.30** |
| | 6 | 82.13 | 45.51 |

(d) LWM-Text-Chat-128k on Multi-LexSum

| Method | $\gamma$ | Acceptance Rate ↑ | Speedup ↑ |
|---|---|---|---|
| StreamingLLM | 1 | 83.96 | **37.80** |
| | 2 | 77.41 | 35.13 |
| | 3 | 71.28 | 32.37 |
| SnapKV | 1 | 89.25 | **39.04** |
| | 2 | 83.53 | 37.22 |
| | 3 | 80.04 | 35.48 |
| QuantSpec | 1 | 95.94 | 42.79 |
| | 2 | 95.06 | 47.10 |
| | 4 | 92.55 | 48.15 |
| | 6 | 87.73 | **48.20** |

# H. Comparing Acceptance Rates

Although the acceptance rates in Table 4 for the sparse KV baselines and QuantSpec do not seem exceedingly different, this is misleading because the acceptance rates shown are for the optimal $\gamma$ values observed from our hyperparameter search in Table 6. Table 4 effectively compares the acceptance rates of the baselines at very low $\gamma$ (e.g. 1) with those of QuantSpec at much higher $\gamma$ (e.g. 6). Here, we compare the acceptance rates of different self-speculative decoding algorithms. For fair comparison, Figure 9 illustrates the acceptance rate between the draft and target models as a function of speculation length. We observe that QuantSpec consistently outperforms sparse KV approaches in terms of acceptance rate. Notably, as speculation length increases, the acceptance rate of sparse KV methods degrades much faster, whereas our method maintains high acceptance rates.

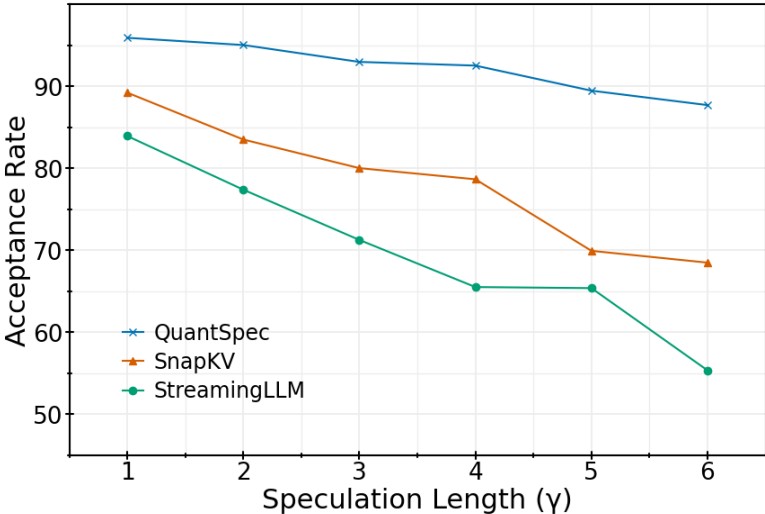

*Figure 9.* Acceptance rate of self-speculative decoding methods at different speculation length measured for model LWM-Text-Chat-128k on Multi-LexSum dataset.

## I. Additional Results

Here we present some additional results on newer model architectures. Table 7 shows acceptance rate and speedup evaluations on Mistral and Llama 3.1 models. The results follow the same trends and conclusions as the results in the main paper.

*Table 7.* Efficiency result of QuantSpec on **Mistral-v0.3** and **Llama 3.1** models on the Multi-LexSum dataset for different context lengths on a single RTX A6000 GPU. We adopt the best speculative length $\gamma$ for each method.

| **Mistral-7B-v0.3** | | | | |
|---|---|---|---|---|
| Context Length | Method | Acceptance Rate ↑ (optimal $\gamma$) | Peak GPU Memory (GB) ↓ | Speedup (× AR) ↑ |
| 16k | StreamingLLM | 0.86 (1) | 16.25 | 0.94 |
| | SnapKV | 0.86 (1) | 16.70 | 0.93 |
| | QuantSpec | 0.94 (6) | 17.42 | **1.55** |
| 32k | StreamingLLM | 0.89 (1) | 18.67 | 1.07 |
| | SnapKV | 0.85 (1) | 19.79 | 0.98 |
| | QuantSpec | 0.93 (6) | 18.34 | **1.61** |
| **Llama-3.1-8B** | | | | |
| Context Length | Method | Acceptance Rate ↑ (optimal $\gamma$) | Peak GPU Memory (GB) ↓ | Speedup (× AR) ↑ |
| 16k | StreamingLLM | 0.63 (1) | 17.73 | 0.82 |
| | SnapKV | 0.66 (1) | 18.18 | 0.85 |
| | QuantSpec | 0.90 (6) | 18.79 | **1.48** |
| 32k | StreamingLLM | 0.81 (1) | 20.15 | 0.93 |
| | SnapKV | 0.73 (1) | 21.27 | 0.90 |
| | QuantSpec | 0.92 (6) | 19.82 | **1.54** |
| 128k | StreamingLLM | 0.89 (1) | 34.91 | 1.06 |
| | SnapKV | 0.80 (1) | 39.95 | 1.06 |
| | QuantSpec | 0.91 (6) | 26.05 | **1.63** |

# J. Kernel Implementation Detail

In our approach, we implement the algorithm using a Flash Decoding framework. In the initial stage of the Flash Decoding process, we compute the log-sum-exp (LSE) values over the INT4-quantized key-value (KV) cache. To facilitate parallelism, the keys and values are partitioned into smaller chunks, with each chunk length set as a multiple of the quantization group size. This segmentation enables parallel computation of attention between the query and each chunk. During this process, the LSE values for individual chunks are recorded.

For the draft model, we begin by loading only the upper 4 bits of the INT4-quantized KV cache within each chunk, along with the corresponding scaling factors and zero points. These are then dequantized in the kernel to reconstruct the KV cache, and the LSE is computed following the standard Flash Decoding procedure. During the verification phase, both the upper and lower bits of the INT4-quantized KV cache are loaded and dequantized. Simultaneously, a separate computation of the LSE is performed using the full-precision KV cache retained in BF16 format. Since the residual cache length exceeds the speculative length, the attention computation over the quantized region is inherently non-causal. Consequently, attention masking is applied only to the full-precision segment.

Finally, in the second stage of the Flash Decoding algorithm, the LSE values obtained from both the quantized and BF16 segments are merged to form an integrated representation that captures information from the complete KV cache.

---

**Algorithm 1** QuantSpec Algorithm

---

**Input:** Model $M$, Upper 4-bit Cache $C_U$, Lower 4-bit Cache $C_L$, Full Precision Cache Buffer $[C_{F_1}, C_{F_2}]$,

**Input:** Prefill Length $S_P$, Target Decode Length $S_D$, Prefill Context $X = [x_0, \cdots, x_{S_P-1}]$, Speculate Length $\gamma$

**Input:** Number of Layers $L$, Sensitive Layer Number $L_S$, Quantization Group Size $G$

**Function:** PREFILL, DRAFT, TARGET, VERITY, QUANTIZE, REJECTCACHE

**Notation:** Verified tokens $x_i$, generated draft tokens $g_i$, logits of draft model $q_i$, logits of target model $p_i$, number of tokens already been generated in total $N$, number of tokens already been generated in this speculate step $n$, number of tokens accepted in this speculate step $v$

**Prefill Stage**

1: $X_{S_P}, C_{KV} \leftarrow \text{PREFILL}(M, X_{<S_P})$                ▷ KV Cache is written together for simplicity

2: $C_U, C_L \leftarrow \text{QUANTIZE}(C_{KV:S_P-G}, L_S)$           ▷ Prepare the hierarchical quantized KV Cache

3: $C_{F_1}, C_{F_2} \leftarrow C_{KV_{S_P-G:}}, \text{None}$           ▷ Prepare the full-precision cache buffer for recent tokens

**Decode Stage**

4: **while** $N < S_D$ **do**

5:     $n \leftarrow 0$ and $v \leftarrow 0$

6:     **while** $n < \gamma$ **do**

7:        $q_{n+1}, C_{F_2} \leftarrow \text{DRAFT}(M, C_U, C_{F_1}, C_{F_2}, L_S, X_{\leq S_P+N} + g_{<n})$

8:        Sample $g_{n+1} \sim q_{n+1}$ and $n \leftarrow n + 1$

9:     **end while**

10:    $p_1, \cdots p_\gamma, C_{F_2} \leftarrow \text{TARGET}(M, C_U, C_L, C_{F_1}, C_{F_2}, X_{\leq S_P+N} + g_{<\gamma})$

11:    **for** $i = 1$ to $\gamma$ **do**

12:       **if** $\text{VERIFY}(g_i, p_i, q_i)$ **then**

13:          $x_{N+i} \leftarrow g_i$ and $v \leftarrow v + 1$

14:       **else**

15:          $x_{N+i} \leftarrow \text{CORRECT}(p_i, q_i)$ and $v \leftarrow v + 1$

16:          $C_{F_2} \leftarrow \text{REJECTCACHE}(C_{F_2}, i)$      ▷ Clear the rejected KV cache from the full precision cache buffer

            Break

17:       **end if**

18:       **if** $i = \gamma$ **then**

19:          $x_{N+\gamma+1} \leftarrow p_{\gamma+1}$ and $N \leftarrow N + 1$

20:       **end if**

21:    **end for**

      $N \leftarrow N + v$

22:    **if** $C_{F_2}$ is full **then**

23:       Concatenate $\text{QUANTIZE}(C_{F_1})$ with $C_U$ and $C_L$     ▷ Quantize the first half of the full precision cache buffer

24:       $C_{F_1} \leftarrow C_{F_2:-G}$ and $C_{F_2} \leftarrow C_{F_2-G:}$     ▷ Move the second part to the first part of full precision buffer

25:    **end if**

26: **end while**

---

