# OpenReview forum: "QuantSpec: Self-Speculative Decoding with Hierarchical Quantized KV Cache"
_ICML.cc/2025/Conference — ICML 2025 poster_

### Official Review · Reviewer_ce5e · 2025-03-07

**Overall Recommendation:** 3

**Summary:**

This paper introduces QuantSpec, a self-speculative decoding framework designed specifically for long-context LLM inference. The framework employs a draft model that shares the architecture of the original model but implements hierarchical 4-bit quantized KV cache and 4-bit quantized weights for acceleration. QuantSpec achieves up to 2.5x end-to-end speedup while reducing memory usage by 1.3x compared to other self-speculative decoding methods for long-context scenarios.

## update after rebuttal
I raised my score to 3 since the authors addressed most of my questions.

**Claims And Evidence:**

Yes, the evidence presented adequately supports the claims proposed in the paper.

**Essential References Not Discussed:**

No essential references appear to be missing from the discussion.

**Experimental Designs Or Analyses:**

The experimental analysis is generally well-executed. However, the experimental design has several notable issues:

1. The paper states in its abstract that KV-cache has been the main bottleneck in long-context LLM inference for edge devices. However, evaluations were conducted on an A5000 GPU, which is not an edge device but rather a professional-grade GPU.
2. The tested models are outdated. Llama-2-7b was released two years ago, and demonstrating that the method works on this model has limited relevance to current applications. Contemporary models have different characteristics that may affect the method's efficacy.
3. The paper lacks comparison with MagicDec, which is a significant omission given its relevance to the self-speculative decoding approach.

**Methods And Evaluation Criteria:**

The method is technically sound but lacks significant novelty in its approach.

**Other Comments Or Suggestions:**

No.

**Other Strengths And Weaknesses:**

Strengths:

1. The paper is well-written with clear presentation that enhances readability and comprehension.
2. The authors demonstrate significant technical effort through the implementation of high-performance custom CUDA kernels, which is commendable.

**Questions For Authors:**

I have two primary questions regarding the evaluation settings:

1. Llama-2-7b is relatively outdated at this point. Why didn't you evaluate your approach on more recent models that better represent the current state of the field? Including contemporary models would significantly strengthen your claims about the method's broad applicability.
2. Your paper lacks comparison with MagicDec, which employs a similar self-speculative approach. What advantages does your method offer over MagicDec specifically? This comparison seems essential for properly positioning your contribution in the literature.

**Relation To Broader Scientific Literature:**

The idea of using the model itself for speculative decoding is not novel, as similar approaches have been implemented in prior works such as MagicDec and TriForce. Additionally, the application of 4-bit hierarchical KV cache represents an incremental improvement rather than a substantial innovation in the field.

**Theoretical Claims:**

No issues were found in the theoretical claims presented.

---

> ### Author Rebuttal · Authors · 2025-04-01
>
> Thank you for your valuable comments. We appreciate that you find the paper technically sound and well-written. We address your questions and comments in detail below:
>
> > R4-1: Evaluate your approach on more recent models that better represent the current state of the field.
>
> **A**: We appreciate the reviewer’s feedback on including newer models. Below we show results on **Mistral-v0.3** and **Llama 3.1** models on the Multi-LexSum dataset for different context lengths on a single RTX A6000 GPU. We adopt the best speculative length γ for each method. These results follow the same trends and conclusions as the results in the main paper. We will add these results and more in the Appendix.
>
>
> ### Mistral-7B-v0.3
>
> |Context Length|Method|Acceptance Rate ↑ (optimal γ)|Peak GPU Memory (GB) ↓|Speedup (× AR) ↑|
> |:-------------:|:----:|:---------------------------:|:--------------------:|:--------------:|
> |16k|StreamingLLM|0.86 (1)|16.25|0.94|
> ||SnapKV|0.86 (1)|16.70|0.93|
> ||**QuantSpec**|0.94 (6)|17.42|**1.55**|
> |32k|StreamingLLM|0.89 (1)|18.67|1.07|
> ||SnapKV|0.85 (1)|19.79|0.98|
> ||**QuantSpec**|0.93 (6)|18.34|**1.61**|
>
> ### Llama-3.1-8B
>
> |Context Length|Method|Acceptance Rate ↑ (optimal γ)|Peak GPU Memory (GB) ↓|Speedup (× AR) ↑|
> |:-------------:|:----:|:---------------------------:|:--------------------:|:--------------:|
> |16k|StreamingLLM|0.63 (1)|17.73|0.82|
> ||SnapKV|0.66 (1)|18.18|0.85|
> ||**QuantSpec**|**0.90 (6)**|18.79|**1.48**|
> |32k|StreamingLLM|0.81 (1)|20.15|0.93|
> ||SnapKV|0.73 (1)|21.27|0.90|
> ||**QuantSpec**|0.92 (6)|19.82|**1.54**|
> |128k|StreamingLLM|0.89 (1)|34.91|1.06|
> ||SnapKV|0.80 (1)|39.95|1.06|
> ||**QuantSpec**|0.91 (6)|26.05|**1.63**|
>
>
> ---
>
> > R4-2: Your paper lacks a comparison with MagicDec, which employs a similar self-speculative approach.
>
> **A**: Thank you for pointing this out. We would like to clarify that the baselines we adopt in our paper, that we referred as StreamingLLM and SnapKV, are in fact the two variants introduced by MagicDec [1]. As such, our experiments already provide a direct empirical comparison with the methods proposed in MagicDec. We show that our approach outperforms these baselines in efficiency, demonstrating the effectiveness of our method. We will clarify this more explicitly in the revised version.
>
> ---
>
> > R4-3: What advantages does your method offer over MagicDec specifically?
>
> **A**: For long context settings, MagicDec employs self-speculative decoding with sparse KV cache for draft model which leads to poor acceptance rates especially in tasks where full context is important e.g. summarization. In contrast, our method employs KV cache quantization to accelerate draft models which **leads to better acceptance rate and thus much better speedups** (please refer to Results section and Appendix H for exact numbers). Additionally, due to the hierarchical nature of quantization, we enable bit sharing between the target and draft KV cache, which **saves GPU memory**. However for the sparse KV methods used by MagicDec, draft model and target model must use a separate copy of KV Cache, leading to higher memory consumption.
>
> ---
>
> > R4-4: The paper states in its abstract that KV-cache has been the main bottleneck in long-context LLM inference for edge devices. However, evaluations were conducted on an A6000 GPU, which is not an edge device but rather a professional-grade GPU.
>
> **A**: Yes that is definitely right that A6000 GPU is not an edge device. However, we kindly note that edge hardware often have worse memory bandwidth to compute ratio which actually makes KV Cache bottleneck an even bigger problem. For instance, the Nvidia Jetson AGX Orin 64GB, a typical edge device, provides up to 204 GB/s of memory bandwidth and 275 TOPS [2] of compute performance. In comparison, the RTX A6000 offers a higher memory bandwidth of 768 GB/s and 309.7 TFLOPS [3] . Notably, the memory-to-compute ratio on the Jetson AGX Orin is lower than that of the A6000, making it more susceptible to memory bandwidth limitations. As a result, **the speedup benefits of QuantSpec by reducing KV cache size are expected to be even more pronounced on edge devices**, where memory bandwidth is a more significant bottleneck. In the final version of the paper we will add an experiment with an edge hardware to showcase this along with a discussion on this.
>
> [1] MagicDec: Breaking the Latency-Throughput Tradeoff for Long Context Generation with Speculative Decoding, ICLR 2025
>
> [2] https://www.nvidia.com/content/dam/en-zz/Solutions/gtcf21/jetson-orin/nvidia-jetson-agx-orin-technical-brief.pdf
>
> [3] https://www.nvidia.com/content/dam/en-zz/Solutions/design-visualization/quadro-product-literature/proviz-print-nvidia-rtx-a6000-datasheet-us-nvidia-1454980-r9-web%20(1).pdf

---

### Official Review · Reviewer_5aMm · 2025-03-11

**Overall Recommendation:** 3

**Summary:**

In long context senarios, loading KV cache is a major bottleneck in both memory and latency. This paper introduces QuantSpec, which is a self-speculative decoding framework designed to accelerate long-context inference. Unlike existing speculative decoding methods that using a smaller model as the draft model, QuantSpec uses the same model with 4-bit quantized weights as the draft model, along with the hierarchical 4-bit quantized KV cache. This approach enables end-to-end speedups of up to 2.5×.

**Claims And Evidence:**

As listed near the end of introduction, the main claims made in the submission are (I skipped the roofline analysis here):

(1) A new hierarchical quantization technique that enables bit-sharing between the target and draft models’ KV caches

(2) A double full-precision cache buffer used for storing the most recent KV cache in full precision to improve acceptance rates and also reduce the number of quant. and dequant. operations

(3) Provide custom CUDA kernels for attention with hierarchical quantized KV cache achieving up to 2.88× speedups at 4-bit precision relative to FP16 FlashAttention kernels

All three claims are verified empirically in the experiment section

**Essential References Not Discussed:**

NA

**Experimental Designs Or Analyses:**

Yes, I checked the validity of the experimental  analyses. It makes sense to me

**Methods And Evaluation Criteria:**

Yes, the applied benchmark dataset makes sense for the problem and application.

**Other Comments Or Suggestions:**

NA

**Other Strengths And Weaknesses:**

NA

**Questions For Authors:**

Can the author elaborate more details on the CUDA kernel design?

**Relation To Broader Scientific Literature:**

Speed up long context generation is a very important problem. As more and more people rely on chatbot to process long documents.

**Theoretical Claims:**

There is no theoretical claims in this paper

---

> ### Author Rebuttal · Authors · 2025-04-01
>
> We thank the reviewer for the valuable comments. We address the questions below:
>
> > R3-1: Can the author elaborate more details on the CUDA kernel design?
>
> **A**: We will add a section in Appendix outlining our kernel design. We include a short summary here for the reviewer:
>
> In our approach, we implement the algorithm using a Flash Decoding framework. In the initial stage of the Flash Decoding process, we compute the log-sum-exp (LSE) values over the INT4-quantized key-value (KV) cache. To facilitate parallelism, the keys and values are partitioned into smaller chunks, with each chunk length set as a multiple of the quantization group size. This segmentation enables parallel computation of attention between the query and each chunk. During this process, the LSE values for individual chunks are recorded.
>
> For the draft model, we begin by loading only the upper 4 bits of the INT4-quantized KV cache within each chunk, along with the corresponding scaling factors and zero points. These are then dequantized in the kernel to reconstruct the KV cache, and the LSE is computed following the standard Flash Decoding procedure. During the verification phase, both the upper and lower bits of the INT4-quantized KV cache are loaded and dequantized. Simultaneously, a separate computation of the LSE is performed using the full-precision KV cache retained in BF16 format. Since the residual cache length exceeds the speculative length, the attention computation over the quantized region is inherently non-causal. Consequently, attention masking is applied only to the full-precision segment.
>
> Finally, in the second stage of the Flash Decoding algorithm, the LSE values obtained from both the quantized and BF16 segments are merged to form an integrated representation that captures information from the complete KV cache.

---

### Official Review · Reviewer_srdL · 2025-03-14

**Overall Recommendation:** 4

**Summary:**

The authors introduce a novel speculative decoding based technique for speeding up LLM inference. The basic idea is to use a hierarchical quantized KV cache and quantize model weights instead of storing a separate KV cache for both the target model and the draft model and storing separate model weights.The basic technical Insight is that you can decompose an int8 value into two int4 components which motivates a hierarchical KV cache structure and also allows for the shared architecture between the draft and target model.

## post rebuttal
Thanks for your thorough response! I'll update my score; your response was very thorough.

**Claims And Evidence:**

The claims are all supported by two sets of experiments. QuantSpec achieves speedups against streamingllm and snapKV, supporting the design choices of using quantize weights and cache rather than a sparse KV cache. The author is also provide insights into regimes in which quantizing weights versus the KV cache perform better. I also appreciated the analysis on using customized Cuda kernels.

**Essential References Not Discussed:**

NA

**Experimental Designs Or Analyses:**

I did check the soundness and validity of experimental designs. I don't have any issues to point out.

**Methods And Evaluation Criteria:**

I think the evaluations and baselines make sense. SnapKV and streaming LM weren't necessarily designed for the purposes of being used in a speculative decoding framework but I think that makes them reasonable baselines.

**Other Comments Or Suggestions:**

N/A

**Other Strengths And Weaknesses:**

N/A

**Questions For Authors:**

Do you have intuition on why key and value caches have distinct quantization strategies?

**Relation To Broader Scientific Literature:**

The paper addresses a well-known technical problem (e.g,. speeding up inference in llms). I think the paper motivates why naive approaches to speculative decoding will fail in this regime and also provides interesting insights on how the draft model should be instantiated

**Theoretical Claims:**

NA

---

> ### Author Rebuttal · Authors · 2025-04-01
>
> Thank you for your valuable comments. We are happy that you found the insights from the paper interesting. We address your questions in detail below:
>
> > R2-1: Do you have intuition on why key and value caches have distinct quantization strategies?
>
> **A**: The KV cache in transformer models serve distinct purposes, which inform their optimal quantization strategies. Key vectors are reused across tokens and accessed along the channel dimension, making **per-channel quantization** better for preserving directional consistency. In contrast, value vectors are consumed per token and contribute to weighted outputs, making **per-token quantization** more effective. We follow a previous work KIVI [1] which also leverages this asymmetry design to apply a 2-bit quantization that reduces memory and achieves a relatively good performance.
>
> [1] KIVI: A Tuning-Free Asymmetric 2bit Quantization for KV Cache, ICML 2024

---

### Official Review · Reviewer_ikhW · 2025-03-14

**Overall Recommendation:** 3

**Summary:**

This submission introduces QuantSpec, a novel self-speculative decoding method that employs KV-Cache quantisation during token drafting, to optimise the inference efficiency on long-context LLMs. Based on the insight that in long-context sequences much of the pressure in memory bandwidth is attributed to loading KV Cache entries during decoding, QuantSpec introduces a hierachical 4-8 bit quantized KV cache, used for drafting and verification respectively. A custom CUDA kernel implementation tailored to the proposed KV cache structure allows the proposed approach to realise inference speed-up in real-world GPU deployment.

**Claims And Evidence:**

The claims made in this submission are supported by clear evidence for the examined cases. However, a broader analysis is required to solidify the findings and demonstrate the generality of the proposed approach. Please see below for more details suggestions.

**Essential References Not Discussed:**

The manuscript adequately cites relevant work in general. Particularly for the adopted quantization scheme, a comparative discussion with the approach of AnyPrecisionLLM [ICML'24] (which solely focuses on weights, however is also hierarchical and thus relevant to the proposed method) is required for completeness.

**Experimental Designs Or Analyses:**

The presented experiments are convincing and demonstrate the effectiveness of the proposed approach via real-world deployments measurements, in the examined use-cases.

**Methods And Evaluation Criteria:**

The proposed method and evaluation criteria are meaningful and suit well the examined problem and usecases.

**Other Comments Or Suggestions:**

POST REBUTTAL EDIT: Provisionally increasing my score from 2 to 3, having read the thorough replies of the authors to all raised comments.

**Other Strengths And Weaknesses:**

Strengths:
- Overall the manuscript studies a very interesting and timely problem.
- The provided discussion and analysis offers numerous insights for the efficient deployment of LLMs.
- The proposed approach is simple and effective and suits well the examined scenario, while minimizing any deployment overheads.
- The proposed double cache buffer and custom CUDA kernel allows the proposed methodology to yield speed-ups, realisable during deployment on commodity GPUs.

Comments:
- It is unclear whether the findings of the analysis in Fig.2 and corresponding results in Fig.4 are representative of real world use-cases. Although the provided results study a wide range of realistic context lengths, the size of the model (Llama2-7B) is disproportional to the typical size of LLMs used for long context inference, which usually approach or surpass 1T params. Please see questions below too.
- Some of the parameters of the analysis remain vague and need to be discussed in more detail. For example what is considered "small" or "large" batch in the discussion of Sec. 3.1.2.
- It is unclear whether the proposed methodology can also be applied on the more general self-speculative decoding setting, where the draft model is a subset of the original (verification LLM), extracted e.g. through pruning (SWIFT [ICLR'25]) or early-exiting (LayerSkip [ACL'24]).

**Questions For Authors:**

1. How do the findings of Fig.2 and 4 scale with the parameter count of the backbone LLM ? Long-context LLMs typically adopt larger model backbones, which may again be bound by weight transfers from memory. Are there any representative works of 7B models adopting >100k token context length?
2. How do the finding of Sec.3 change with batch size of 1? What is defined as a "small" and "large" batch in the discussion?
3. How is the proposed quantisation scheme positioned to other nested quantisation approaches, typically applied in weights, as in AnyPrecision LLM? What is the main benefit of the proposed scheme compared to these works?
4. Can the proposed be applied in more traditional self-speculative decoding, where the draft model is a subset of the verification one e.g. SWIFT or LayerSkip ?

**Relation To Broader Scientific Literature:**

The key contributions of the manuscript focuses on improving the inference efficiency of self-speculative decoding in long-context LLM inference. The proposed approach is based on KV Cache quantisation during drafting, with minimal overheads and respecting the assumptions of the original inference scheme.

**Theoretical Claims:**

Not Applicable.

---

> ### Author Rebuttal · Authors · 2025-04-01
>
> Thank you for your valuable comments. We are happy that you find discussion and analysis in the paper insightful. We address your questions and comments in detail below:
>
> > R1-1: How do the findings of Fig.2 and 4 scale with the parameter count of the backbone LLM ? Long-context LLMs typically adopt larger model backbones, which may again be bound by weight transfers from memory.
>
> **A**: This is a great question. The findings still hold even for larger model backbones for long context lengths. To be specific, the minimum context length for which KV cache starts becoming the main bottleneck increases linearly with active parameter count (in case of MoE, active parameters can be much less than the total parameter count). However, with increasing batch size KV cache again starts to become bottleneck even for shorter context lengths. Below, we have included the exact formula to check when KV cache becomes the main bottleneck. Also it is worth noting that our method uses **weight quantization** for draft model to provide reliable speedups even for short contexts lengths where weight transfers are the main bottleneck.
>
> KV cache dominates when
> $\text{KV cache elements} \approx 2 \cdot L \cdot B \cdot S \cdot d_{\text{model}} > \text{Model Weights} $, where $L$ is the number of layers, $B$ is the batch size, $S$ is the sequence length and $d_{model}$ is the hidden dim.
>
> ---
>
> > R1-2: Are there any representative works of 7B models adopting >100k token context length?
>
> **A**: Yes, Qwen offers a 7B model (Qwen2.5-7B-Instruct-1M) trained with a context length of 1M tokens. Other examples include Meta’s Llama-3.1-8B model and Gemma 3 4B which were both trained with a context length of 128K. Moreover, given recent inference-time scaling research, the demand for small models with long context lengths is further increasing. This is because the long reasoning traces outputted by the model becomes part of its context, so model providers are increasingly extending the context lengths of small reasoning models to outperform larger models. Thus, long context use cases are important even for smaller (<7B) language models.
>
> ---
>
> > R1-3: How do findings of Section 3 change with batch size of 1, what is defined as a "small" and "large" batch in the discussion?
>
> **A**: We define small batch sizes as 1-8, and large batch sizes as greater than 32. The findings of Section 3 show that for small batch sizes (including batch size of 1), short contexts (1-8k) benefit from weight quantization, medium contexts (8k-64k) benefit from both weight and KV cache quantization, and long contexts (>64k) benefit from KV cache-only quantization. We will clarify this in the final version.
>
> ---
>
> > R1-4: How does your quantization scheme for KV cache different from other nested quantization approaches like AnyPrecision LLM.
>
> **A**: AnyPrecision LLM [1] truncates the 8-bit representation to get the corresponding upper 4-bit version. However, truncation introduces bias in the 4-bit quantization case, leading to higher quantization error of the upper 4-bit cache and lower acceptance rates. Our approach reduces this bias by first quantizing the upper 4-bit, then quantizing the residual part to get the lower 4-bit. This results in a more accurate quantization and a better acceptance rate.  Below we show the quantization error on a random fp16 tensor of length 100. We also report acceptance rate we get when using Anyprecision vs our quantization scheme. For this analysis, we used Llama-2-7B-32K-Instruct model on Multi-LexSum dataset with a prefill length of 32k.
>
> | Method                    | Quantization Error (MSE) ↓ | Acceptance Rate ↑ |
> |:-------------------:|:--------------------:|:------------------:|
> | Anyprecision 4 bit  |        0.761                                  |       0.86                     |
> | QuantSpec  4 bit    |        0.013                                  |       0.92                     |
>
> We will add this analysis in Appendix and will also add a discussion about nested quantization in related works.
>
> ---
>
> > R1-5:  Can your method be applied with traditional self-speculative decoding approaches like SWIFT or LayerSkip?
>
> **A**: This is a very interesting point. QuantSpec can be applied on top of other traditional self-speculative decoding methods like LayerSkip [2] and SWIFT [3]. For example, the algorithm proposed in SWIFT could be used to identify important and unimportant layers, and QuantSpec could then be used to retain important layers in higher precision.
> We will add a discussion about compatibility with other self-speculative decoding algorithms in the Appendix.
>
> ---
>
>
> [1] Any-Precision LLM: Low-Cost Deployment of Multiple, Different-Sized LLMs, ICML 2024
>
> [2] LayerSkip: Enabling Early Exit Inference and Self-Speculative Decoding, ACL 2024
>
> [3] SWIFT: On-the-Fly Self-Speculative Decoding for LLM Inference Acceleration, ICLR 2025

---

> > ### Comment · Reviewer_ikhW · 2025-04-05
> >
> > I would like to thank the authors for thoroughly addressing all raised comments in their rebuttal. Having read their replies, as well as the other reviewers' comments, I am inclined to provisionally increase my score; ahead of the upcoming discussion between reviewers.
> >
> > I would also like to emphasize, that although it makes sense to consider batch sizes 1-8 as "small", batch size =1 vs batch size >=2 can imply fundamentally different application domains, e.g. it is rarely possible to find applications for on-device deployment of LLMs with batch size greater than 1. As such, I believe that the batch size =1 should be discussed and evaluated independently, throughout the paper.

---

> > > ### Author Response · Authors · 2025-04-05
> > >
> > > We appreciate the reviewer checking all the responses. We agree that practically batch size 1 vs >1 can imply different domains and we will make sure to update the analysis section in the paper to stress on this point.

---

### Decision · Program_Chairs · 2025-05-01

**Decision:**

Accept (poster)

**Comment:**

Summary:

QuantSpec is a novel self-speculative decoding framework designed for efficient long-context LLM inference. Recognizing that memory bandwidth bottlenecks primarily arise from loading large KV caches, QuantSpec employs a hierarchical KV-cache quantization strategy, using 4-bit quantization for token drafting and higher precision (8-bit) for verification. Unlike conventional speculative decoding methods relying on smaller auxiliary models, QuantSpec leverages the same model architecture with 4-bit quantized weights for drafting, which enables sharing between draft and target models. A custom CUDA kernel implementation further optimizes real-world GPU deployments.

Strengths:

1. The proposed method is simple and effective. The authors develop custom CUDA kernels for the proposed method, which makes the experiments much more convincing.

2. All reviewers agree that the problem of study is important and timely.

3. The provided discussion and analysis offers numerous insights for the efficient deployment of LLMs.

Weaknesses:

1. The authors are encouraged to discuss the case of bs=1 in more detail, as bs=1 and bs>1 are two different scenarios for applications.

2. As the motivation of this work is on edge device, the authors are encouaged to run experiments on real edge devices instead of professional-grade A5000 GPUs.

3. Some missing experiments are suggested by Reviewer ce5e.

All reviewers vote for (weak) acceptance. AC agrees with reviewers' recommendation and suggest weak accept for this paper.